# Maximization of Approximately Submodular Functions

**Thibaut Horel**
Harvard University
thorel@seas.harvard.edu

**Yaron Singer**
Harvard University
yaron@seas.harvard.edu

## Abstract

We study the problem of maximizing a function that is *approximately* submodular under a cardinality constraint. Approximate submodularity implicitly appears in a wide range of applications as in many cases errors in evaluation of a submodular function break submodularity. Say that $F$ is $\varepsilon$-approximately submodular if there exists a submodular function $f$ such that $(1-\varepsilon)f(S) \le F(S) \le (1+\varepsilon)f(S)$ for all subsets $S$. We are interested in characterizing the query-complexity of maximizing $F$ subject to a cardinality constraint $k$ as a function of the *error level* $\varepsilon > 0$. We provide both lower and upper bounds: for $\varepsilon > n^{-1/2}$ we show an exponential query-complexity lower bound. In contrast, when $\varepsilon < 1/k$ or under a stronger *bounded curvature* assumption, we give constant approximation algorithms.

## 1 Introduction

In recent years, there has been a surge of interest in machine learning methods that involve discrete optimization. In this realm, the evolving theory of *submodular optimization* has been a catalyst for progress in extraordinarily varied application areas. Examples include active learning and experimental design [9, 12, 14, 19, 20], sparse reconstruction [1, 6, 7], graph inference [23, 24, 8], video analysis [29], clustering [10], document summarization [21], object detection [27], information retrieval [28], network inference [23, 24], and information diffusion in networks [17].

The power of submodularity as a modeling tool lies in its ability to capture interesting application domains while maintaining provable guarantees for optimization. The guarantees however, apply to the case in which one has access to the exact function to optimize. In many applications, one does not have access to the exact version of the function, but rather some approximate version of it. If the approximate version remains submodular then the theory of submodular optimization clearly applies and modest errors translate to modest loss in quality of approximation. But if the approximate version of the function ceases to be submodular all bets are off.

**Approximate submodularity.** Recall that a function $f : 2^N \to \mathbb{R}$ is *submodular* if for all $S, T \subseteq N$, $f(S \cup T) + f(S \cap T) \le f(S) + f(T)$. We say that a function $F : 2^N \to \mathbb{R}$ is $\varepsilon$-*approximately submodular* if there exists a submodular function $f : 2^N \to \mathbb{R}$ s.t. for any $S \subseteq N$:

$$(1 - \varepsilon)f(S) \le F(S) \le (1 + \varepsilon)f(S). \tag{1}$$

Unless otherwise stated, all submodular functions $f$ considered are normalized ($f(\emptyset) = 0$) and *monotone* ($f(S) \le f(T)$ for $S \subseteq T$). Approximate submodularity appears in various domains.

- **Optimization with noisy oracles.** In these scenarios, we wish to solve optimization problems where one does not have access to a submodular function but a noisy version of it. An example recently studied in [5] involves maximizing information gain in graphical models; this captures many Bayesian experimental design settings.

- **PMAC learning.** In the active area of learning submodular functions initiated by Balcan and Harvey [3], the objective is to *approximately* learn submodular functions. Roughly speaking, the PMAC-learning framework guarantees that the learned function is a constant-factor approximation of the true submodular function with high probability. Therefore, after learning a submodular function, one obtains an approximately submodular function.

- **Sketching.** Since submodular functions have, in general, exponential-size representation, [2] studied the problem of *sketching* submodular functions: finding a function with polynomial-size representation approximating a given submodular function. The resulting sketch is an approximately submodular function.

**Optimization of approximate submodularity.** We focus on optimization problems of the form

$$\max_{S \,:\, |S| \le k} F(S) \tag{2}$$

where $F$ is an $\varepsilon$-approximately submodular function and $k \in \mathbb{N}$ is the cardinality constraint. We say that a set $S \subseteq N$ is an $\alpha$-approximation to the optimal solution of (2) if $|S| \le k$ and $F(S) \ge \alpha \max_{|T| \le k} F(T)$. As is common in submodular optimization, we assume the *value query model*: optimization algorithms have access to the objective function $F$ in a black-box manner, *i.e.* they make queries to an oracle which returns, for a queried set $S$, the value $F(S)$. The query-complexity of the algorithm is simply the number of queries made to the oracle. An algorithm is called an $\alpha$-approximation algorithm if for any approximately submodular input $F$ the solution returned by the algorithm is an $\alpha$-approximately optimal solution. Note that if there exists an $\alpha$-approximation algorithm for the problem of maximizing an $\varepsilon$-approximate submodular function $F$, then this algorithm is a $\frac{\alpha(1-\varepsilon)}{1+\varepsilon}$-approximation algorithm for the original submodular function $f$[1]. Conversely, if no such algorithm exists, this implies an inapproximability for the original function.

Clearly, if a function is 0-approximately submodular then it retains desirable provable guarantees[2], and it it is arbitrarily far from being submodular it can be shown to be trivially inapproximable (*e.g.* maximize a function which takes value of 1 for a single arbitrary set $S \subseteq N$ and 0 elsewhere). The question is therefore:

> *How close should a function be to submodular to retain provable approximation guarantees?*

In recent work, it was shown that for any constant $\varepsilon > 0$ there exists a class of $\varepsilon$-approximately submodular functions for which no algorithm using fewer than exponentially-many queries has a constant approximation ratio for the canonical problem of maximizing a monotone submodular function under a cardinality constraint [13]. Such an impossibility result suggests two natural relaxations: the first is to make additional assumptions about the structure of errors, such a stochastic error model. This is the direction taken in [13], where the main result shows that when errors are drawn i.i.d. from a wide class of distributions, optimal guarantees are obtainable. The second alternative is to assume the error is subconstant, which is the focus of this paper.

## 1.1 Overview of the results

Our main result is a spoiler: even for $\varepsilon = 1/n^{1/2-\beta}$ for any constant $\beta > 0$ and $n = |N|$, no algorithm can obtain a constant-factor approximation guarantee. More specifically, we show that:

- For the general case of **monotone submodular functions**, for any $\beta > 0$, given access to a $\frac{1}{n^{1/2-\beta}}$-approximately submodular function, no algorithm can obtain an approximation ratio better than $O(1/n^\beta)$ using polynomially many queries (Theorem 3);

- For the case of **coverage functions** we show that for any fixed $\beta > 0$ given access to an $\frac{1}{n^{1/3-\beta}}$-approximately submodular function, no algorithm can obtain an approximation ratio strictly better than $O(1/n^\beta)$ using polynomially many queries (Theorem 4).

The above results imply that even in cases where the objective function is arbitrarily close to being submodular as the number $n$ of elements in $N$ grows, reasonable optimization guarantees are unachievable. The second result shows that this is the case even when we aim to optimize *coverage functions*. Coverage functions are an important class of submodular functions which are used in numerous applications [11, 21, 18].

**Approximation guarantees.** The inapproximability results follow from two properties of the model: the structure of the function (submodularity), and the size of $\varepsilon$ in the definition of approximate submodularity. A natural question is whether one can relax either conditions to obtain positive approximation guarantees. We show that this is indeed the case:

- In the general case of **monotone submodular functions** we show that the greedy algorithm achieves a $\left(1-1/e-O(\delta)\right)$ approximation ratio when $\varepsilon = \frac{\delta}{k}$ (Theorem 5). Furthermore, this bound is tight: given a $1/k^{1-\beta}$-approximately submodular function, the greedy algorithm no longer provides a constant factor approximation guarantee (Proposition 6).
- Since our query-complexity lower bound holds for coverage functions, which already contain a great deal of structure, we relax the structural assumption by considering functions with **bounded curvature** $c$; this is a common assumption in applications of submodularity to machine learning and has been used in prior work to obtain theoretical guarantees [15, 16]. Under this assumption, we give an algorithm which achieves an approximation ratio of $(1 - c)(\frac{1-\varepsilon}{1+\varepsilon})^2$ (Proposition 8).

We state our positive results for the case of a cardinality constraint of $k$. Similar results hold for matroids of rank $k$, the proofs of those can be found in the Appendix. Note that cardinality constraints are a special case of matroid constraints, therefore our lower bounds also apply to matroid constraints.

## 1.2 Discussion and additional related work

Before transitioning to the technical results, we briefly survey error in applications of submodularity and the implications of our results to these applications. First, notice that there is a coupling between approximate submodularity and erroneous evaluations of a submodular function: if one can evaluate a submodular function within (multiplicative) accuracy of $1 \pm \varepsilon$ then this is an $\varepsilon$-*approximately submodular* function.

**Additive vs multiplicative approximation.** The definition of approximate submodularity in (1) uses relative (multiplicative) approximation. We could instead consider absolute (additive) approximation, *i.e.* require that $f(S) - \varepsilon \leq F(S) \leq f(S) + \varepsilon$ for all sets $S$. This definition has been used in the related problem of optimizing approximately convex functions [4, 25], where functions are assumed to have normalized range. For un-normalized functions or functions whose range is unknown, a relative approximation is more informative. When the range is known, specifically if an upper bound $B$ on $f(S)$ is known, an $\varepsilon/B$-approximately submodular function is also an $\varepsilon$-additively approximate submodular function. This implies that our lower bounds and approximation results could equivalently be expressed for additive approximations of normalized functions.

**Error vs noise.** If we interpret Equation (1) in terms of error, we see that no assumption is made on the source of the error yielding the approximately submodular function. In particular, there is no stochastic assumption: the error is deterministic and worst-case. Previous work have considered submodular or combinatorial optimization under random noise. Two models naturally arise:

- *consistent noise:* the approximate function $F$ is such that $F(S) = \xi_S f(S)$ where $\xi_S$ is drawn independently for each set $S$ from a distribution $\mathcal{D}$. The key aspect of consistent noise is that the random draws occur only once: querying the same set multiple times always returns the same value. This definition is the one adopted in [13]; a similar notion is called *persistent noise* in [5].
- *inconsistent noise:* in this model $F(S)$ is a random variable such that $f(S) = \mathbb{E}[F(S)]$. The noisy oracle can be queried multiple times and each query corresponds to a new independent random draw from the distribution of $F(S)$. This model was considered in [26] in the context of dataset summarization and is also implicitly present in [17] where the objective function is defined as an expectation and has to be estimated via sampling.

Formal guarantees for consistent noise have been obtained in [13]. A standard way to approach optimization with inconsistent noise is to estimate the value of each set used by the algorithm to an accuracy $\varepsilon$ via independent randomized sampling, where $\varepsilon$ is chosen small enough so as to obtain approximation guarantees. Specifically, assuming that the algorithm only makes polynomially many value queries and that the function $f$ is such that $F(S) \in [b, B]$ for any set $S$, then a classical application of the Chernoff bound combined with a union bound implies that if the value of each set is estimated by averaging the value of $m$ samples with $m = \Omega\left(\frac{B \log n}{b \varepsilon^2}\right)$, then with high probability the estimated value $F(S)$ of each set used by the algorithm is such that $(1 - \varepsilon)f(S) \leq F(S) \leq (1 + \varepsilon)f(S)$. In other words, *randomized sampling is used to construct a function which is $\varepsilon$-approximately submodular with high probability.*

**Implications of results in this paper.** Given the above discussion, our results can be interpreted in the context of noise as providing guarantees on what is a tolerable noise level. In particular, Theorem 5 implies that if a submodular function is estimated using $m$ samples, with $m = \Omega\left(\frac{B n^2 \log n}{b}\right)$, then the Greedy algorithm is a constant approximation algorithm for the problem of maximizing a monotone submodular function under a cardinality constraint. Theorem 3 implies that if $m = O\left(\frac{B n \log n}{b}\right)$ then the resulting estimation error is within the range where no algorithm can obtain a constant approximation ratio.

## 2 Query-complexity lower bounds

In this section we give query-complexity lower bounds for the problem of maximizing an $\varepsilon$-approximately submodular function subject to a cardinality constraint. In Section 2.1, we show an exponential query-complexity lower bound for the case of general submodular functions when $\varepsilon \geq n^{-1/2}$ (Theorem 3). The same lower-bound is then shown to hold even when we restrict ourselves to the case of coverage functions for $\varepsilon \geq n^{-1/3}$ (Theorem 4).

**A general overview of query-complexity lower bounds.** At a high level, the lower bounds are constructed as follows. We define a class of monotone submodular functions $\mathcal{F}$, and draw a function $f$ uniformly at random from $\mathcal{F}$. In addition we define a submodular function $g : 2^N \to \mathbb{R}$ s.t. $\max_{|S| \leq k} f(S) \geq \rho(n) \cdot \max_{|S| \leq k} g(S)$, where $\rho(n) = o(1)$ for a particular choice of $k < n$. We then define the approximately submodular function $F$:

$$F(S) = \begin{cases} g(S), & \text{if } (1 - \varepsilon)f(S) \leq g(S) \leq (1 + \varepsilon)f(S) \\ f(S) & \text{otherwise} \end{cases}$$

Note that by its definition, this function is an $\varepsilon$-approximately submodular function. To show the lower bound, we reduce the problem of proving inapproximability of optimizing an approximately submodular function to the problem of distinguishing between $f$ and $g$ using $F$. We show that for every algorithm, there exists a function $f \in \mathcal{F}$ s.t. if $f$ is unknown to the algorithm, it cannot distinguish between the case in which the underlying function is $f$ and the case in which the underlying function is $g$ using polynomially-many value queries to $F$, even when $g$ is known to the algorithm. Since $\max_{|S| \leq k} f(S) \geq \rho(n) \max_{|S| \leq k} g(S)$, this implies that no algorithm can obtain an approximation better than $\rho(n)$ using polynomially-many queries; otherwise such an algorithm could be used to distinguish between $f$ and $g$.

### 2.1 Monotone submodular functions

**Constructing a class of hard functions.** A natural candidate for a class of functions $\mathcal{F}$ and a function $g$ satisfying the properties described in the overview is:

$$f^H(S) = |S \cap H| \quad \text{and} \quad g(S) = \frac{|S|h}{n}$$

for $H \subseteq N$ of size $h$. The reason why $g$ is hard to distinguish from $f^H$ is that when $H$ is drawn uniformly at random among sets of size $h$, $f^H$ is close to $g$ with high probability. This follows from an application of the Chernoff bound for negatively associated random variables. Formally, this is stated in Lemma 1 whose proof is given in the Appendix.

**Lemma 1.** *Let $H \subseteq N$ be a set drawn uniformly among sets of size $h$, then for any $S \subseteq N$, writing $\mu = \frac{|S|h}{n}$, for any $\varepsilon$ such that $\varepsilon^2 \mu > 1$:*

$$\mathbb{P}_H\big[(1-\varepsilon)\mu \leq |S \cap H| \leq (1+\varepsilon)\mu\big] \geq 1 - 2^{-\Omega(\varepsilon^2 \mu)}$$

Unfortunately this construction fails if the algorithm is allowed to evaluate the approximately submodular function at small sets: for those the concentration of Lemma 1 is not high enough. Our construction instead relies on designing $\mathcal{F}$ and $g$ such that when $S$ is "large", then we can make use of the concentration result of Lemma 1 and when $S$ is "small", functions in $\mathcal{F}$ and $g$ are deterministically close to each other. Specifically, we introduce for $H \subseteq N$ of size $h$:

$$f^H(S) = |S \cap H| + \min\left(|S \cap (N \setminus H)|, \alpha\left(1 - \frac{h}{n}\right)\right)$$

$$g(S) = \min\left(|S|, \frac{|S|h}{n} + \alpha\left(1 - \frac{h}{n}\right)\right)$$

(3)

The value of the parameters $\alpha$ and $h$ will be set later in the analysis. Observe that when $S$ is small ($|S \cap \bar{H}| \leq \alpha(1 - h/n)$ and $|S| \leq \alpha$) then $f^H(S) = g(S) = |S|$. When $S$ is large, Lemma 1 implies that $|S \cap H|$ is close to $|S|h/n$ and $|S \cap (N \setminus H)|$ is close to $|S|(1 - h/n)$ with high probability.

First note that $f^H$ and $g$ are monotone submodular functions. $f^H$ is the sum of a monotone additive function and a monotone budget-additive function. The function $g$ can be written $g(S) = G(|S|)$ where $G(x) = \min(x, xh/n + \alpha(1 - h/n))$. $G$ is a non-decreasing concave function (minimum between two non-decreasing linear functions) hence $g$ is monotone submodular.

Next, we observe that there is a gap between the maxima of the functions $f^H$ and the one of $g$. When $S \leq k$, $g(S) = \frac{|S|h}{n} + \alpha\left(1 - \frac{h}{n}\right)$. The maximum is clearly attained when $|S| = k$ and is upper-bounded by $\frac{kh}{n} + \alpha$. For $f^H$, the maximum is equal to $k$ and is attained when $S$ is a subset of $H$ of size $k$. So for $\alpha \leq k \leq h$, we obtain:

$$\max_{|S| \leq k} g(S) \leq \left(\frac{\alpha}{k} + \frac{h}{n}\right) \max_{|S| \leq k} f^H(S), \quad H \subseteq N$$

(4)

**Indistinguishability.** The main challenge is now to prove that $f^H$ is close to $g$ with high probability. Formally, we have the following lemma.

**Lemma 2.** *For $h \leq \frac{n}{2}$, let $H$ be drawn uniformly at random among sets of size $h$, then for any $S$:*

$$\mathbb{P}_H\big[(1-\varepsilon)f^H(S) \leq g(S) \leq (1+\varepsilon)f^H(S)\big] \geq 1 - 2^{-\Omega(\varepsilon^2 \alpha h/n)}$$

(5)

*Proof.* For concision we define $\bar{H} := N \setminus H$, the complement of $H$ in $N$. We consider four cases depending on the cardinality of $S$ and $S \cap \bar{H}$.

**Case 1:** $|S| \leq \alpha$ and $|S \cap \bar{H}| \leq \alpha\left(1 - \frac{h}{n}\right)$. In this case $f^H(S) = |S \cap H| + |S \cap \bar{H}| = |S|$ and $g(S) = |S|$. The two functions are equal and the inequality is immediately satisfied.

**Case 2:** $|S| \leq \alpha$ and $|S \cap \bar{H}| \geq \alpha(1 - \frac{h}{n})$. In this case $g(S) = |S| = |S \cap H| + |S \cap \bar{H}|$ and $f^H(S) = |S \cap H| + \alpha(1 - \frac{h}{n})$. By assumption on $|S \cap \bar{H}|$, we have:

$$(1-\varepsilon)\alpha\left(1 - \frac{h}{n}\right) \leq |S \cap \bar{H}|$$

For the other side, by assumption on $|S \cap \bar{H}|$, we have that $|S| \geq \alpha(1 - \frac{h}{n}) \geq \frac{\alpha}{2}$ (since $h \leq \frac{n}{2}$). We can then apply Lemma 1 and obtain:

$$\mathbb{P}_H\left[|S \cap \bar{H}| \leq (1+\varepsilon)\alpha\left(1 - \frac{h}{n}\right)\right] \geq 1 - 2^{-\Omega(\varepsilon^2 \alpha h/n)}$$

**Case 3:** $|S| \geq \alpha$ and $|S \cap \bar{H}| \geq \alpha\left(1 - \frac{h}{n}\right)$. In this case $f^H(S) = |S \cap H| + \alpha(1 - \frac{h}{n})$ and $g(S) = \frac{|S|h}{n} + \alpha(1 - \frac{h}{n})$. We need to show that:

$$\mathbb{P}_H\left[(1-\varepsilon)\frac{|S|h}{n} \leq |S \cap H| \leq (1+\varepsilon)\frac{|S|h}{n}\right] \geq 1 - 2^{-\Omega(\varepsilon^2 \alpha h/n)}$$

This is a direct consequence of Lemma 1.

**Case 4:** $|S| \geq \alpha$ and $|S \cap \bar{H}| \leq \alpha\left(1 - \frac{h}{n}\right)$. In this case $f^H(S) = |S \cap H| + |S \cap \bar{H}|$ and $g(S) = \frac{|S|h}{n} + \alpha(1 - \frac{h}{n})$. As in the previous case, we have:

$$\mathbb{P}_H\left[(1 - \varepsilon)\frac{|S|h}{n} \leq |S \cap H| \leq (1 + \varepsilon)\frac{|S|h}{n}\right] \geq 1 - 2^{-\Omega(\varepsilon^2 \alpha h/n)}$$

By the assumption on $|S \cap \bar{H}|$, we also have:

$$|S \cap \bar{H}| \leq \alpha\left(1 - \frac{h}{n}\right) \leq (1 + \varepsilon)\alpha\left(1 - \frac{h}{n}\right)$$

So we need to show that:

$$\mathbb{P}_H\left[(1 - \varepsilon)\alpha\left(1 - \frac{h}{n}\right) \leq |S \cap \bar{H}|\right] \geq 1 - 2^{-\Omega(\varepsilon^2 \alpha h/n)}$$

and then we will be able to conclude by union bound. This is again a consequence of Lemma 1. $\square$

**Theorem 3.** *For any $0 < \beta < \frac{1}{2}$, $\varepsilon \geq \frac{1}{n^{1/2-\beta}}$, and any (possibly randomized) algorithm with query-complexity smaller than $2^{\Omega(n^{\beta/2})}$, there exists an $\varepsilon$-approximately submodular function $F$ such that for the problem of maximizing $F$ under a cardinality constraint, the algorithm achieves an approximation ratio upper-bounded by $\frac{2}{n^{\beta/2}}$ with probability at least $1 - \frac{1}{2^{\Omega(n^{\beta/2})}}$.*

*Proof.* We set $k = h = n^{1-\beta/2}$ and $\alpha = n^{1-\beta}$. Let $H$ be drawn uniformly at random among sets of size $h$ and let $f^H$ and $g$ be as in (3). We first define the $\varepsilon$-approximately submodular function $F^H$:

$$F^H(S) = \begin{cases} g(S) & \text{if } (1 - \varepsilon)f^H(S) \leq g(S) \leq (1 + \varepsilon)f^H(S) \\ f^H(S) & \text{otherwise} \end{cases}$$

It is clear from the definition that this is an $\varepsilon$-approximately submodular function. Consider a deterministic algorithm $A$ and let us denote by $S_1, \ldots, S_m$ the queries made by the algorithm when given as input the function $g$ ($g$ is 0-approximately submodular, hence it is a valid input to $A$). Without loss of generality, we can include the set returned by the algorithm in the queries, so $S_m$ denotes the set returned by the algorithm. By (5), for any $i \in [m]$:

$$\mathbb{P}_H[(1 - \varepsilon)f^H(S_i) \leq g(S_i) \leq (1 + \varepsilon)f^H(S_i)] \geq 1 - 2^{-\Omega\left(n^{\frac{\beta}{2}}\right)}$$

when these events realize, we have $F^H(S_i) = g(S_i)$. By union bound over $i$, when $m < 2^{\Omega\left(n^{\frac{\beta}{2}}\right)}$:

$$\mathbb{P}_H[\forall i, F^H(S_i) = g(S_i)] > 1 - m2^{-\Omega\left(n^{\beta/2}\right)} = 1 - 2^{-\Omega\left(n^{\beta/2}\right)} > 0$$

This implies the existence of $H$ such that $A$ follows the same query path when given $g$ and $F^H$ as inputs. For this $H$:

$$F^H(S_m) = g(S_m) \leq \max_{|S| \leq k} g(S) \leq \left(\frac{\alpha}{k} + \frac{h}{n}\right) \max_{|S| \leq k} f^H(S) = \left(\frac{\alpha}{k} + \frac{h}{n}\right) \max_{|S| \leq k} F^H(S)$$

where the second inequality comes from (4). For our choice of parameters, $\frac{\alpha}{k} + \frac{h}{n} = 2/n^{\frac{\beta}{2}}$, hence:

$$F^H(S_m) \leq \frac{2}{n^{\frac{\beta}{2}}} \max_{|S| \leq k} F^H(S)$$

Let us now consider the case where the algorithm $A$ is randomized and let us denote $A_{H,R}$ the solution returned by the algorithm when given function $F^H$ as input and random bits $R$. We have:

$$\mathbb{P}_{H,R}\left[F^H(A_{H,R}) \leq \frac{2}{n^{\beta/2}} \max_{|S| \leq k} F^H(S)\right] = \sum_r \mathbb{P}[R = r]\mathbb{P}_H\left[F^H(A_{H,R}) \leq \frac{2}{n^{\beta/2}} \max_{|S| \leq k} F^H(S)\right]$$

$$\geq (1 - 2^{-\Omega(n^{\frac{\beta}{2}})}) \sum_r \mathbb{P}[R = r] = 1 - 2^{-\Omega(n^{\beta}2)}$$

where the equality comes from the analysis of the deterministic case (when the random bits are fixed, the algorithm is deterministic). This implies the existence of $H$ such that:

$$\mathbb{P}_R\left[F^H(A_{H,R}) \leq \frac{2}{n^{\beta/2}} \max_{|S| \leq k} F^H(S)\right] \geq 1 - 2^{-\Omega(n^{\beta}2)}$$

and concludes the proof of the theorem. $\square$

## 2.2 Coverage functions

In this section, we show that an exponential query-complexity lower bound still holds even in the restricted case where the objective function approximates a coverage function. Recall that by definition of a coverage function, the elements of the ground set $N$ are subsets of a set $\mathcal{U}$ called the *universe*. For a set $S = \{\mathcal{S}_1, \ldots, \mathcal{S}_m\}$ of subsets of $\mathcal{U}$, the value $f(S)$ is given by $f(S) = |\bigcup_{i=1}^m \mathcal{S}_i|$.

**Theorem 4.** *For any $0 < \beta < \frac{1}{2}$, $\varepsilon \geq \frac{1}{n^{1/3-\beta}}$, and any (possibly randomized) algorithm with query-complexity smaller than $2^{\Omega(n^{\beta/2})}$, there exists a function $F$ which $\varepsilon$-approximates a coverage function, such that for the problem of maximizing $F$ under a cardinality constraint, the algorithm achieves an approximation ratio upper-bounded by $\frac{2}{n^{\beta/2}}$ with probability at least $1 - \frac{1}{2^{\Omega(n^{\beta/2})}}$.*

The proof of Theorem 4 has the same structure as the proof of Theorem 3. The main difference is a different choice of class of functions $\mathcal{F}$ and function $g$. The details can be found in the appendix.

# 3 Approximation algorithms

The results from Section 2 can be seen as a strong impossibility result since an exponential query-complexity lower bound holds even in the specific case of coverage functions which exhibit a lot of structure. Faced with such an impossibility, we analyze two ways to relax the assumptions in order to obtain positive results. One relaxation considers $\varepsilon$-approximate submodularity when $\varepsilon \leq \frac{1}{k}$; in this case we show that the Greedy algorithm achieves a constant approximation ratio (and that $\varepsilon = \frac{1}{k}$ is tight for the Greedy algorithm). The other relaxation considers functions with stronger structural properties, namely, functions with *bounded curvature*. In this case, we show that a constant approximation ratio can be obtained for any constant $\varepsilon$.

## 3.1 Greedy algorithm

For the general class of monotone submodular functions, the result of [22] shows that a simple greedy algorithm achieves an approximation ratio of $1 - \frac{1}{e}$. Running the same algorithm for an $\varepsilon$-approximately submodular function results in a constant approximation ratio when $\varepsilon \leq \frac{1}{k}$. The detailed description of the algorithm can be found in the appendix.

**Theorem 5.** *Let $F$ be an $\varepsilon$-approximately submodular function, then the set $S$ returned by the greedy algorithm satisfies:*

$$F(S) \geq \frac{1}{1 + \frac{4k\varepsilon}{(1-\varepsilon)^2}} \left( 1 - \left( \frac{1-\varepsilon}{1+\varepsilon} \right)^{2k} \left( 1 - \frac{1}{k} \right)^k \right) \max_{S:|S| \leq k} F(S)$$

*In particular, for $k \geq 2$, any constant $0 \leq \delta < 1$ and $\varepsilon = \frac{\delta}{k}$, this approximation ratio is constant and lower-bounded by $\left( 1 - \frac{1}{e} - 16\delta \right)$.*

*Proof.* Let us denote by $O$ an optimal solution to $\max_{S:|S| \leq K} F(S)$ and by $f$ a submodular representative of $F$. Let us write $S = \{e_1, \ldots, e_\ell\}$ the set returned by the greedy algorithm and define $S_i = \{e_1, \ldots, e_i\}$, then:

$$f(O) \leq f(S_i) + \sum_{e \in \text{OPT}} \left[ f(S_i \cup \{e\}) - f(S_i) \right] \leq f(S_i) + \sum_{e \in O} \left[ \frac{1}{1-\varepsilon} F(S_i \cup \{e\}) - f(S_i) \right]$$

$$\leq f(S_i) + \sum_{e \in O} \left[ \frac{1}{1-\varepsilon} F(S_{i+1}) - f(S_i) \right] \leq f(S_i) + \sum_{e \in O} \left[ \frac{1+\varepsilon}{1-\varepsilon} f(S_{i+1}) - f(S_i) \right]$$

$$\leq f(S_i) + K \left[ \frac{1+\varepsilon}{1-\varepsilon} f(S_{i+1}) - f(S_i) \right]$$

where the first inequality uses submodularity, the second uses the definition of approximate submodularity, the third uses the definition of the Algorithm, the fourth uses approximate submodularity again and the last one uses that $|O| \leq k$.

Reordering the terms, and expressing the inequality in terms of $F$ (using the definition of approximate submodularity) gives:

$$F(S_{i+1}) \geq \left(1 - \frac{1}{k}\right)\left(\frac{1-\varepsilon}{1+\varepsilon}\right)^2 F(S_i) + \frac{1}{k}\left(\frac{1-\varepsilon}{1+\varepsilon}\right)^2 F(O)$$

This is an inductive inequality of the form $u_{i+1} \geq au_i + b$, $u_0 = 0$. Whose solution is $u_i \geq \frac{b}{1-a}(1 - a^i)$. For our specific $a$ and $b$, we obtain:

$$F(S) \geq \frac{1}{1 + \frac{4k\varepsilon}{(1-\varepsilon)^2}}\left(1 - \left(1 - \frac{1}{k}\right)^k\left(\frac{1-\varepsilon}{1+\varepsilon}\right)^{2k}\right)F(O) \qquad \square$$

The following proposition shows that $\varepsilon = \frac{1}{k}$ is tight for the greedy algorithm, and that this is the case even for additive functions. The proof can be found in the Appendix.

**Proposition 6.** *For any $\beta > 0$, there exists an $\varepsilon$-approximately additive function with $\varepsilon = \Omega\left(\frac{1}{k^{1-\beta}}\right)$ for which the Greedy algorithm has non-constant approximation ratio.*

**Matroid constraint.** Theorem 5 can be generalized to the case of matroid constraints. We are now looking at a problem of the form: $\max_{S \in I} F(S)$, where $I$ is the set of independent sets of a matroid.

**Theorem 7.** *Let $I$ be the set of independent sets of a matroid of rank $k$, and let $F$ be an $\varepsilon$-approximately submodular function, then if $S$ is the set returned by the greedy algorithm:*

$$F(S) \geq \frac{1}{2}\left(\frac{1-\varepsilon}{1+\varepsilon}\right)\frac{1}{1 + \frac{k\varepsilon}{1-\varepsilon}}\max_{S \in I} f(S)$$

*In particular, for $k \geq 2$, any constant $0 \leq \delta < 1$ and $\varepsilon = \frac{\delta}{k}$, this approximation ratio is constant and lower-bounded by $(\frac{1}{2} - 2\delta)$.*

## 3.2 Bounded curvature

With an additional assumption on the curvature of the submodular function $f$, it is possible to obtain a constant approximation ratio for any $\varepsilon$-approximately submodular function with constant $\varepsilon$. Recall that the curvature $c$ of function $f : 2^N \to \mathbb{R}$ is defined by $c = 1 - \min_{a \in N}\frac{f_{N \setminus \{a\}}(a)}{f(a)}$. A consequence of this definition when $f$ is submodular is that for any $S \subseteq N$ and $a \in N \setminus S$ we have that $f_S(a) \geq (1-c)f(a)$.

**Proposition 8.** *For the problem $\max_{|S| \leq k} F(S)$ where $F$ is an $\varepsilon$-approximately submodular function which approximates a monotone submodular $f$ with curvature $c$, there exists a polynomial time algorithm which achieves an approximation ratio of $(1-c)(\frac{1-\varepsilon}{1+\varepsilon})^2$.*

## Footnotes

[1]Observe that for an approximately submodular function $F$, there exists many submodular functions $f$ of which it is an approximation. All such submodular functions $f$ are called *representatives* of $F$. The conversion between an approximation guarantee for $F$ and an approximation guarantee for a representative $f$ of $F$ holds for any choice of the representative.

[2]Specifically, [22] shows that it possible to obtain a $(1 - 1/e)$ approximation ratio for a cardinality constraint.

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
