[Supplementary Material]

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

 3 but uses a different construction for $f^H$ and $g$ since the ones defined defined in Section 2.1 are not coverage functions. For $H \subseteq N$ of size $h$, we define:

$$f^H(S) = \begin{cases} |S \cap H| + \alpha & \text{if } S \neq \emptyset \\ 0 & \text{otherwise} \end{cases} \quad \text{and} \quad g(S) = \begin{cases} \frac{|S|h}{n} + \alpha & \text{if } S \neq \emptyset \\ 0 & \text{otherwise} \end{cases}$$

It is clear that $f^H$ and $g$ can be realized as coverage functions: $|S \cap H|$ and $\frac{|S|h}{n}$ are additive functions which are a subclass of coverage functions. The offset of $\alpha$ can be obtained by having all sets defining $f^H$ and $g$ cover the same $\alpha$ elements of the universe.

We now relate the maxima of $g$ and $f^H$: the maximum of $f^H$ is attained when $S$ is a subset of $H$ of size $k$ and is equal to $k + \alpha \geq k$. The value of $g$ only depends on $|S|$ and is equal to $\frac{kh}{n} + \alpha$ when $|S|$ is of size $k$. Hence:

$$\max_{|S| \leq k} g(S) \leq \left( \frac{\alpha}{k} + \frac{h}{n} \right) \max_{|S| \leq k} f^H(S) \tag{6}$$

We now show a concentration result similar to (4): let $H$ be drawn uniformly at random among sets of size $h$, then for any $S$ and $0 < \varepsilon < 1$:

$$\mathbb{P}_H\left[ (1 - \varepsilon)f^H(S) \leq g(S) \leq (1 + \varepsilon)f^H(S) \right] \geq 1 - 2^{-\Omega(\varepsilon^3 \alpha h/n)} \tag{7}$$

We will consider two cases depending on the size of $|S|$. When $|S| \leq \varepsilon \alpha$, the inequality is deterministic. For the right-hand side:

$$(1 + \varepsilon)f(S) \geq (1 + \varepsilon)\alpha \geq \alpha + |S| \geq \alpha + \frac{|S|h}{n} = g(S)$$

where the first inequality used $|S \cap H| \geq 0$, the second inequality used the bound on $|S|$ and the last inequality used $h \leq n$. For the left-hand side:

$$(1 - \varepsilon)f(S) = (1 - \varepsilon)\alpha + (1 - \varepsilon)|S \cap H| \leq \alpha - \varepsilon\alpha + |S| \leq \alpha \leq g(S)$$

where the first inequality used $1 - \varepsilon \leq 1$ and $|S \cap H| \leq |S|$ and the second inequality used the bound on $|S|$.

Let us now consider the case where $|S| \geq \varepsilon \alpha$. This case follows directly by applying Lemma 1 after observing that when $|S| \geq \alpha, \mu \geq \frac{\varepsilon \alpha h}{n}$.

We can now conclude the proof of Theorem 4 by combining (6) and (7) the exact same manner as the proof of Theorem 3 after setting $h = k = n^{1-\beta/2}$ and $\alpha = n^{1-\beta}$.                    □

## B   Proof of Lemma 1

The Chernoff bound stated in Lemma 1 does not follow from the standard Chernoff bound for independent variables. However, we use the fact that the Chernoff bound also holds under the weaker *negative association* assumption.

**Definition 9.** *Random variables $X_1, \ldots, X_n$ are* negatively associated *iff for every $I \subseteq [n]$ and every non-decreasing functions $f : \mathbb{R}^I \to \mathbb{R}$ and $g : \mathbb{R}^{\bar{I}} \to \mathbb{R}$:*

$$\mathbb{E}[f(X_i, i \in I)g(X_j, j \in \bar{I})] \leq \mathbb{E}[f(X_i, i \in I)]\mathbb{E}[g(X_j, j \in \bar{I})]$$

**Claim 10** ([9]). *Let $X_1, \ldots, X_n$ be $n$ negatively associated random variables taking value in $[0, 1]$. Denote by $\mu = \sum_{i=1}^n \mathbb{E}[X_i]$ the expected value of their sum, then for any $\delta \in [0, 1]$:*

$$\mathbb{P}\left[ \sum_{i=1}^n X_i > (1 + \delta)\mu \right] \leq e^{-\delta^2 \mu/3}$$

$$\mathbb{P}\left[ \sum_{i=1}^n X_i < (1 - \delta)\mu \right] \leq e^{-\delta^2 \mu/2}$$

**Claim 11** ([9]). *Let $H$ be a random subset of size $h$ of $[n]$ and let us define the random variables $X_i = 1$ if $i \in H$ and $X_i = 0$ otherwise. Then $X_1, \ldots, X_n$ are negatively associated.*

The proof of Lemma 1 is now immediate after observing that $|S \cap H|$ can be written $|S \cap H| = \sum_{i \in S} X_i$ where $X_i$ is defined as in Claim 11. Since $\mathbb{P}[X_i = 1] = \frac{h}{n}$ we have $\mu = \frac{|S|h}{n}$.

## C  Proofs for Section 3.1

The full description of the greedy algorithm used in Theorem 5 can be found in Algorithm 1.

---

**Algorithm 1** APPROXIMATEGREEDY

---

1: **initialize** $S \leftarrow \emptyset$
2: **while** $|S| \leq k$ **do**
3: $\quad$ $S \leftarrow S \cup \operatorname{argmax}_{a \in N \setminus S} F(S \cup \{a\})$.
4: **end while**
5: **return** $S$

---

*Proof of Proposition 6.* Fix $\beta > 0$ and $\varepsilon = \frac{1}{k^{1-\beta}}$. Let us consider an additive function $f$ where the ground set $N$ can be written $N = A \cup B \cup C$ with:

- $A$ is a set of $\frac{1}{2\varepsilon}$ elements of value 2.

- $B$ is a set of $\frac{n}{2} - \frac{1}{4\varepsilon}$ elements of value $\frac{1}{n}$.

- $C$ is a set of $\frac{n}{2} - \frac{1}{4\varepsilon}$ elements of value 1.

We now define the following $\varepsilon$-approximately submodular function $F$:

$$F(S) = \begin{cases} \frac{1}{\varepsilon} & \text{if } S = A \cup \{c\} \text{ with } c \in C \\ f(S) & \text{otherwise} \end{cases}$$

$F$ is an $\varepsilon$-approximately submodular function. Indeed, the only case where $F$ differs from $f$ is when $S = A \cup \{c\}$ with $c \in C$. In this case $F(S) = \frac{1}{\varepsilon} \leq \frac{1}{\varepsilon} + 1 = f(S)$ and:

$$F(S) = \frac{1}{\varepsilon} \geq (1 - \varepsilon)\left(\frac{1}{\varepsilon} + 1\right) = (1 - \varepsilon)f(S)$$

When $\varepsilon < \frac{1}{2}$, the greedy algorithm selects all elements from $A$ and spends the remaining budget on $B$ and obtains a value of $\frac{1}{\varepsilon} + \frac{1}{n}(k - k^{1-\beta}/2) = O(k^{1-\beta})$ when given $F$ as input. However, it is clear that the optimal solution for $F$ is to select all elements in $A$ and spend the remaining budget on $C$ for a value of $\frac{1}{\varepsilon} + (k - k^{1-\beta}/2) = \Omega(k)$. The resulting approximation ratio is $O\left(\frac{1}{k^{\beta}}\right)$ which converges to zero as the budget constraint $k$ grows to infinity. $\qquad\square$

Theorem 7 uses a slight modification of Algorithm 1 to accommodate the matroid constraint. The full description is given in Algorithm 2.

---

**Algorithm 2** MATROIDGREEDY

---

1: **initialize** $S \leftarrow \emptyset$
2: **while** $N \neq \emptyset$ **do**
3: $\quad$ $x^* \leftarrow \operatorname{argmax}_{x \in N} F(S \cup \{x\})$
4: $\quad$ **if** $S \cup \{x\} \in I$ **then**
5: $\quad\quad$ $S \leftarrow S \cup \{x\}$
6: $\quad$ **end if**
7: $\quad$ $N \leftarrow N \setminus \{x^*\}$
8: **end while**
9: **return** $S$

---

*Proof of Theorem 7.* Let us consider $S^* \in \text{argmax}_{S \in I} f(S)$. W.l.o.g. we can assume that $S^*$ is a basis of the matroid ($|S^*| = k$). It is clear that the set $S$ returned by Algorithm 2 is also a basis. By the basis exchange property of matroids, there exists $\phi : S^* \to S$ such that:

$$S - \phi(x) + x \in I, \quad x \in S^*$$

Let us write $S^* = \{e_1^*, \ldots, e_k^*\}$ and $S = \{e_1, \ldots, e_k\}$ where $e_i = \phi(e_i^*)$ and define $S_i = \{e_1, \ldots, e_i\}$ then:

$$f(S^*) \leq f(S) + \sum_{i=1}^{k} f_S(e_i^*) \leq f(S) + \sum_{i=1}^{k} f_{S_{i-1}}(e_i^*)$$

$$\leq f(S) + \sum_{i=1}^{k} \left[ \frac{1+\varepsilon}{1-\varepsilon} f(S_i) - f(S_{i-1}) \right]$$

$$= f(S) + \sum_{i=1}^{k} [f(S_i) - f(S_{i-1})] + \frac{2\varepsilon}{1-\varepsilon} \sum_{i=1}^{k} f(S_i)$$

$$\leq 2f(S) + \frac{2k\varepsilon}{1-\varepsilon} f(S)$$

where the first two inequalities used submodularity, the third used the definition of an $\varepsilon$-erroneous oracle, and the fourth used monotonicity. The result then follows by applying the definition of $\varepsilon$-approximate submodularity. $\square$

## D   Proofs for Section 3.2

The proof of Proposition 8 follows from Lemma 12 which shows how to construct an additive approximation of $F$.

**Lemma 12.** *Let $F$ be an $\varepsilon$-approximately submodular function which approximates a submodular function $f$ with bounded curvature c. Let $F_a$ be the function defined by $F_a(S) = \sum_{e \in S} F(e)$ then:*

$$\frac{1-\varepsilon}{1+\varepsilon} F(S) \leq F_a(S) \leq \frac{1}{1-c} \frac{1+\varepsilon}{1-\varepsilon} F(S), \quad S \subseteq N$$

*Proof.* For the left-hand side:

$$F_a(S) = \sum_{e \in S} F(e) \geq (1-\varepsilon) \sum_{e \in S} f(e) \geq (1-\varepsilon) f(S) \geq \frac{1-\varepsilon}{1+\varepsilon} F(S)$$

where the first and third inequalities used approximate submodularity and the second inequality used that submodular functions are subadditive.

For the right-hand side, let us enumerate $S = \{e_1, \ldots, e_\ell\}$ and write $S_i = \{e_1, \ldots, e_i\}$ (with $S_0 = \emptyset$ by convention). Then:

$$F_a(S) = \sum_{i=1}^{\ell} F(e_i) \leq (1+\varepsilon) \sum_{i=1}^{\ell} f(e_i) \leq \frac{1+\varepsilon}{1-c} \sum_{i=1}^{\ell} f_{S_{i-1}}(e_i) = \frac{1+\varepsilon}{1-c} f(S) \leq \frac{1}{1-c} \frac{1+\varepsilon}{1-\varepsilon} F(S)$$

where the first and last inequalities used approximate submodularity, and the second inequality used the curvature assumption. $\square$

*Proof of Proposition 8.* Let us denote by $S_a$ a solution to $\max_{|S| \leq k} F_a(S)$ where $F_a$ is defined as in Lemma 12. Since $F_a$ is an additive function, $S_a$ can be found by querying the value query oracle for $F$ at each singleton and selecting the top $k$. The approximation ratio then follows directly from Lemma 12. $\square$