[Reviews · NeurIPS 2016]

Reviewer 1

Summary

This paper considers the problem of maximizing a set function, f, that is an eps approximation to a monotone submodular function, that is there exists a monotone submodular function F such that (1 – eps) f(S) \leq F(S) \leq (1 + eps) f(S) for all sets S, under a cardinality constraint given access to an oracle for evaluating f. While it has been known that for any constant eps and exponential number of calls to the oracle are needed to obtain a constant approximation when F is submodular, this paper furthers the study of this question by providing several new and interesting facts. • An exponential number of queries are needed to obtain better than O(1/n^beta) when eps \geq 1/(n^(1/2 – beta). • An exponential number of queries are needed to obtain better than O(1/n^beta) when eps \geq 1/(n^(1/3 – beta) in the special case where F is a coverage function. • Greedy achieves 1 – 1/eps – O(delta) approx. when eps = delta/ k and there is a cardinality constraint of k. Moreover, they show this is essentially tight characterization of greedy. • A (1-c)(1-eps)^2/(1+eps)^2 algorithm when F has c curvature.

Qualitative Assessment

This paper furthers the study of a natural question in optimization and machine learning, namely that of a maximizing a function that is approximately monotone submodular under a cardinality constraint. They strengthen previous lower bounds providing dimension dependent bounds on the amount of approximation until which an exponential number of queries. Moreover, they provide improved upper bounds on the quality of approximation achieved as dependent upon k and the curvature of F. While the question studied by this paper is natural and possibly important and the writing is clear, it is less clear to what degree the results are novel and required new insight over previous work. There is little discussion of how hard it what to achieve the lower bound or extend previous lower bound work to achieve their bounds. In other words, the paper would benefit from more clearly illustrating whether they provide a novel analysis or lower bound technique, versus simply applying previous machinery. Moreover, the upper bound they achieve for greedy feels somewhat natural and ultimately it is unclear if the paper actually proposes a new algorithm for performing approximate submodular maximization. Finally, it appears that there is a gap between the lower bounds and the upper bounds they provide again, making it difficult to ascertain whether or not the paper simply follows the natural analysis of previous work or provides fundamental new insights. That said, the question of closing this gap is tantalizing, ultimately resulting in this borderline review. Detailed Comments: - Lines 8-10: are the inequalities on epsilon in the wrong direction? - Line 60 - 63: although it was said that F is assumed to be monotone, here I would be clear about whether there result applied to monotone submodular functions. - Lines 90-96: a little more discussion of best known running times under the assumption would be beneficial for comparison. - Line 145: say a little more about how this lower bound compares to previous lower bounds and the techniques they use. - Line 150: In the line after “an general” should be “a general” - Lines 273 – 281: say more about what the algorithm actually is and how compares to other algorithms. - General: discuss the gaps in upper bounds and lower bounds left open by this work.

Confidence in this Review

2-Confident (read it all; understood it all reasonably well)


Reviewer 2

Summary

The paper considers the problem of maximizing a set function that is approximately submodular under a cardinality constraint. This is a well-motivated variant of the well-studied submodular maximization problem with a cardinality constraint. The main contributions of this paper are inapproximability results that show that, even if the function is very close to being submodular, the best approximations achievable are polynomial. The paper also provides approximation algorithms in the very small error regime as well as for functions of bounded curvature.

Qualitative Assessment

The inapproximability results provided in this paper are interesting and useful to know. The techniques used to obtain the results are not novel, they follow a standard approach for showing such hardness results that was developed in previous work, such as that of Svitkina and Fleischer [1]. The paper is generally well-written. [1] Z. Svitkina, L. Fleischer. Submodular approximation: sampling-based algorithms and lower bounds. == post-rebuttal answer== I have read the entire rebuttal. I agree with the authors that their proof has novel and interesting elements beyond the general approach provided in previous work.

Confidence in this Review

2-Confident (read it all; understood it all reasonably well)


Reviewer 3

Summary

The paper considers the problem of maximizing an approximately submodular function subject to a cardinality constraint of picking at most k elements. The particular definition of approximate submodularity being used is: a function F is epsilon-approximately submodular if there is a submodular function f such that (1-epsilon)*f(S) <= F(S) <= (1+epsilon)*f(S) for all sets S. The question is for what value of epsilon (in relation with n and k), it is possible get a good approximation factor. The paper shows a strong negative result: for epsilon > n^{-1/2+beta}, any algorithm with query complexity at most 2^{n^{beta/2}} cannot beat approximation factor 2/n^{beta/2}. Similar result is also shown for coverage function. The hardness result uses examples with large values of k = n^{1-beta/2}. For the comparison between epsilon and k, the paper shows that for epsilon <= 1/k, the greedy algorithm achieves a constant approximation while for epsilon > k^{-1+beta}, the greedy algorithm fails to achieve a constant approximation. The lower bound proof works by considering two submodular function f, g such that the optimal value for f is much larger than g but an algorithm with few queries to the approximation F cannot distinguish if the true submodular function is f or g. The basic idea is to pick a random set H and f(S) = |S intersect H| while g(S) = |S|*|H|/n. This basic construction doesn't work because f and g are sufficiently different for small sets but the paper provides tweaks to that make it work (basically by forcing small sets to agree).

Qualitative Assessment

The paper provides strong hardness results for a natural approximately submodular model and complement them with some limited algorithmic results for the cases where the function has bounded curvature or the error level is small compared with 1/k. The algorithmic results are all based on known algorithms with simple tweaks to known proofs so the contribution here is mostly in the hardness result, which can be seen as showing this particular modeling of approximately submodular functions is too weak to yield good result and a better model is needed. The proof technique for the hardness results is a modification of previous results and it is perhaps a smaller contribution than the statement of the results themselves.

Confidence in this Review

2-Confident (read it all; understood it all reasonably well)


Reviewer 4

Summary

This paper shows that (1) for relatively small errors in a submodular function, no polynomial time algorithm can get a constant approximation to optimal, and (2) for even smaller levels of errors, the standard greedy algorithm can get a constant approximation to optimal.

Qualitative Assessment

Technical quality: This work gives interesting theoretical results showing how much error is tolerable when optimizing approximately submodular functions. The proofs are sound, and the examples show interesting cases where the standard greedy, or even general polynomial time algorithms, will fail. Novelty/originality: The proofs techniques used to show that polynomial time algorithms do not exist for even epsilon trending towards 0 are very interesting, and show a pretty surprising result (in general, it was fairly well known that even small amounts of error would cause problems for the greedy algorithm, but this paper shows that no algorithm with a reasonable runtime can exist). In addition, this paper shows that for very low levels of error, the simple greedy will work, and for bounded curvature, a similar simple algorithm will work. Potential impact or usefulness: I think that the results are interesting and useful for the submodularity community. Clarity and presentation: I found the introduction of the paper to be relatively clear in presenting the main contributions of the paper, but the later sections were a bit difficult to understand. The choice of what to put in the main text, and what to put in the appendix was also somewhat strange to me (for example, several detailed proofs were included in the main text (which were largely algebraic derivations), while the algorithm corresponding to Proposition 8 was entirely omitted from the main text). --------- Minor comments Line 8-9: the inequalities for epsilon appear reversed Line 10: "constant approximation algorithms": this confused me for a bit (it seemed like constant referred to the number of queries) Line 46 and Theorem 5 use S on the left and right-hand-sides for different purposes Line 93: Your algorithm only needs to make n queries! This is even fewer than the greedy algorithm, but you don't even mention this fact in the main text. Similarly, in Proposition 8, why is it not included that the algorithm only queries the singleton sets (ie. n queries)?evince Line 131-132: => "polynomially many value queries" eq (3): f^H(S) might be easier to understand if written as \min(|S|, |S\cap H| + \alpha (1 - h/n)) This might also make the proof of lemma 2 slightly cleaner. Line 183-184: "S is a subset of H of size k and is equal to k" This wording is very confusing.

Confidence in this Review

2-Confident (read it all; understood it all reasonably well)


Reviewer 5

Summary

This paper studies the problem of maximizing \epsilon-approximately submodular functions under cardinality constraints, where "\epsilon-approximately submodularity" means multiplicative approximation of submodularity. Several hardness results are devised, e.g., if \epsilon > 1/\sqrt{n} = o(1), constant approximation using polynomially many queries is impossible. Then the paper investigates when constant approximations are achievable; \epsilon = \delta/k and bounded curvature.

Qualitative Assessment

Technical quality: 3 [proofs] + Proofs are sound. [results] + The paper provides not only a general hardness result (Theorem 3), but also study a special case of coverage functions (Theorem 4) in detail. + The relaxation conditions for which reasonable approximations can be obtained, such as bounded curvature, are also provided (Theorems 5 and 8). Novelty / originality: 4 + The paper introduces a new notion called "\epsilon-approximately submodularity" and discusses the hardness results and approximation guarantees for maximization problems. These results are substantially novel. + The paper thoroughly discusses the differences from the existing results and related notions as in Sec. 1.2, and so it is clear for the readers to understand the standing point and novelty of the work compared to existing work. Potential impact or usefulness: 4 + The paper gives examples of approximate submodularity, i.e., "optimization with noisy oracles", "PAC learning", and "sketching", which are useful when applying to concrete applications. + The theoretical results not only just provide hardness, but also relaxation conditions where constant approximations are achievable, which might be practical. Clarity and presentation: 4 [explanations] Overall, the paper is consistently well-written. + Three examples described in Introduction clearly strengthen the motivation of this study. + Providing an overview of the proofs (Lines 151--159) for hardness results is appropriate for the reader to follow the flow. [typo] - In Line 8 of Abstract, the paper said that "for \epsilon < n^{-1/2} we show an exponential query-complexity lower bound," which seems to be "\epsilon > n^{-1/2}". Similarly, "\epsilon > 1/k" in Line 9 might be "\epsilon < 1/k". More understandable explanations would be recommended.

Confidence in this Review

2-Confident (read it all; understood it all reasonably well)


Reviewer 6

Summary

The paper studies approximation of a function whose values are within (1+ε)-multiplicative error to some submodular function under a cardinality constraint. They proved that if ε > n^{-1/2} then any algorithm with polynomially many queries cannot achieve O(1)-approximation. This lower bound is improved for coverage functions. For the upper bound, they showed that if ε < 1/k or a curvature condition holds then one can obtain O(1)-approximation.

Qualitative Assessment

This paper is well-written and the problem considered there is very natural and important for many applications. Especially the lower bound was surprising because subconstant error rate is not enough! Although the proof is based on the standard argument of query complexity and greedy algorithm, they clarified some technical difficulties. Overall, I think this paper achieves an important step for submodular maximization. Minor comment: in abstract, ε < n^{-1/2} should be ε > n^{-1/2}?

Confidence in this Review

2-Confident (read it all; understood it all reasonably well)